# Visual Harmony of the Proportion of Water and Greenery in Urban Streams: Baxi Stream, Yongan City, China

**Jinn-Chyi Chen** [1,*,†] **, Xun-Rui Fan** [2] **, Jian-Qiang Fan** [1] **, Xi-Zhu Lai** [1] **, Gui-Liang Li** [1] **and Feng-Bin Li** [1]

1 School of Hydraulic Engineering, Fujian College of Water Conservancy and Electric Power, Yongan 366000, China
2 School of Architecture Engineering, Fujian College of Water Conservancy and Electric Power, Yongan 366000, China
* Correspondence: chenjinnchyi@gmail.com
† Previously at Department of Landscape and Environmental Design, Huafan University, New Taipei 223011, Taiwan.

**Abstract:** This study investigated the visual harmony of an urban stream considering changes to the ratio of water to greenery on the riverbed. The Baxi stream, a third-order stream in Yongan City, Fujian Province, China was selected as the study site. The stream reach is disturbed by several hydraulic structures, such as restricted water flow by a vertical revetment and water level regulation by submerged dams. Images of the river were captured, and image processing was performed to change the proportion of water and greenery, and the proportions of various landscape elements in the image were calculated. Based on the statistical analysis of survey results, cognitive indicators (vividness and naturalness) associated with harmony and preference, and the relationship between harmony or preference and landscape elements, were established. Landscape elements included ratios of visible water ($W_R$), visible greenery ($G_R$), visible buildings, and visible infrastructure. The results demonstrated that visual preference, $P$, is positively correlated with harmony, $H$, vividness, $V$, and naturalness, $N$. In particular, $H$ is almost consistent to $P$. The proportion of visible water and greenery had a significant impact on the $H$ and $P$ of the stream landscape. When the ratio of $W_R$ to $G_R$ was approximately 0.8, $H$ was optimal, and the public's $P$ was high. These results can be used to improve and enhance the visual landscape quality of this stream reach. The methodology proposed in this study could provide other study areas with a reference for how to obtain the best visual harmony or achieve public acceptance by changing the amount of visible water and/or greenery.

**Keywords:** visible water; visible greenery; visual harmony; visual preference

## 1. Introduction

The riparian zone generally encompasses a vegetated strip of land that extends along streams and rivers and serves as the interface between terrestrial and aquatic ecosystems [1,2]. Riparian zones can reduce surface runoff and erosion, and provide absorption buffers for sediments and nutrients flowing into the rivers from land [3]. A vegetated riparian zone can improve water quality by providing shade, leaf matter, and wood, as well as stabilizing stream banks [4]. Riparian vegetation maintains biodiversity in the river area, protects the riverbank from the direct impact of floods [5], and enhances the appearance and recreational purpose of the area. In addition, wet riparian soils are generally carbon rich and oxygen poor, facilitating nitrogen loss through denitrification [4]. However, vegetation on the riverbank and riverbed increases flow resistance. Therefore, in order to prevent flooding, vegetation is often removed before the flood season to ensure that water flow is not obstructed. During the dry season, the river level is low, so an adjustable dam is used to raise the water level, with the aim of increasing the aesthetic quality of the riverside landscape by increasing the water volume. However, when the water level rises, the riverside vegetation becomes submerged in water, which can make the landscape unnatural

and inharmonious. Therefore, due to changes in the natural environment (hydrology) or anthropogenic interventions, the proportion of water and greenery in the riparian zone is constantly changing. The proportion of water and greenery affects the aesthetic quality and visual harmony. Aesthetic preference is therefore an important issue in urban river reconstruction and waterfront planning, design, and improvement. Improving aesthetic quality is the focus of this study and the ultimate goal of urban river transformation [6].

*1.1. Human Preferences for River Landscapes*

From an evolutionary perspective, humans typically prefer landscapes with water bodies because water is required for human survival [7]. Semi-open landscapes tend to be more popular than dense or open forest landscapes, because semi-open landscapes are features of the African savanna, which is the environment of human origin [7,8]. Human preferences for river landscapes have been addressed in several studies that have explored, for example, riparian vegetation and river planform, flow discharge [9,10], water quality [11,12], wood in rivers [13,14], sediment control structures in mountain streams [15–17], river recreational infrastructure [18–20], and river rehabilitation and restoration [21,22]. Several tools, methods, and models have been proposed to assess the aesthetic quality of the river landscape [23–28]. Research has demonstrated that humans prefer open water and natural river landscapes, but dislike swampy wetlands, dead wood in rivers, and algae-infested waterways [11,20,22]. Too little water, excess vegetation without proper management, and too many human facilities can reduce the visual quality of the river [17,29]. The public support riparian landscapes that convey a sense of care and cleanliness and are typically like a river environment with a sense of safety [21]. The public does not like or accept poorly maintained urban riverscapes, where poor maintenance can manifest through the presence of trash, weeds, or poor water quality [30,31]. Development of efficient ecological and recreational river functions can help improve public appreciation of river landscapes [21]. Existing literature has demonstrated that there are many factors that affect river landscape preference, including water quantity and quality, channel shape, biophysical elements such as vegetation type and biodiversity, artificial facilities, safety, and recreational function. However, most of the previous studies qualitatively studied the correlation between these parameters and river landscape preference. There is a lack of quantitative research on the relationship between visual harmony, river landscape preference, and the ratio of visible water to the amount of greenery.

*1.2. Research on Harmony*

The term harmony is widely used in many fields including engineering, environment, management, urban spaces, art, and design [29,32–36]. The definition of harmony varies between research fields. Keywords related to harmony in landscape and color include "coherence", "unity", "order", "balance", and "pleasing" [37–40]. Many studies have addressed harmony related to the water environment, including river ecosystems [41–43], water resource management [36,44], river landscapes with hydraulic structures [29], and the relationship between humans and water [45–47]. When studying visual aesthetics or visual quality related to the water landscape, aesthetic preference or public perception is a common evaluation index [17,27,48], because absolute aesthetic quality or beauty are subjective terms that depend on personal experiences [40]. Harmony is an important cognitive indicator for evaluating visual preference [29,38]. In certain environmental conditions, harmony has a high correlation with visual preference [29,40] and high levels of harmony help improve scenic beauty and public acceptance [16,49]. Therefore, some studies have evaluated the visual beauty of river landscapes by estimating harmony [29]. Harmony is a term that is familiar to civil and hydraulic engineers. In ecological engineering of soil and water conservation or mountain river projects, there has been an emphasis on harmony or design that balances engineering and nature [50]. However, there remain few studies on the harmony between river engineering and landscape, and the majority of studies have been conducted from the perspective of visual quality or preference [16,17,25,28].

Chen et al. [29] examined the visual quality and visual harmony of soil and water conservation engineering in mountain streams and proposed relevant evaluation methods. Their studies focused on the harmony of mountain stream landscapes with hydraulic structures. However, there are differences between mountain streams and urban streams in terms of topographic features and the composition of landscape elements. Generally, there is less visible greenery coverage and more visible water in an urban stream than in a mountain stream, and there are different types of hydraulic structures. Check dams with heights > 5 m are commonly used to control sediment in mountain streams, but rarely used in urban streams. Common river crossing structures in urban streams are generally lower in height, such as submerged dams and groundsills. These structures act to regulate water intake and water level, as well as stabilise the riverbed. Hydraulic structures also play an important role in urban-stream beautification. Some hydraulic structures in urban rivers, such as submerged dams in the present study area, are designed to improve the visual quality of the water environment.

In this study, we examined the harmony of the river landscape caused by the proportion of landscape elements, particularly water and greenery. We use harmony to refer to the visual unity, balance, and pleasant arrangement of river landscape elements [51]. River landscape elements are categorized as soft landscapes (such as water and greenery) and hard landscapes (such as hydraulic structures, buildings, and infrastructure). We selected the Baxi stream, a third-order urban stream in Fujian Province, China, as our study area. Adjustable submerged dams were constructed in the Baxi stream and have caused variations in the ratio of water to greenery. Significant changes to the visible water level can be detrimental to the harmony of the landscape. Research is required to address the relative proportions of water and greenery in the riverbed landscape to achieve harmony and gain the public's preference. The objectives of this study are as follows: (1) Examine the relationship between harmony and cognitive factors, such as visual preference, vividness, and naturalness in the Baxi stream and understand the differences between visual landscape quality and hydraulic structures in the mountain and urban streams. (2) Understand the relative proportions of water and greenery required to achieve harmony and gain public preference. Further understand the relationship between the water-to-greenery ratio, harmony, and public preference, and establish relevant empirical equations to provide a reference for improving the river landscape and visual quality for the example study area. (3) Discuss the applicability of the empirical equations proposed in this study.

## 2. Study Area

The study area is located in the Baxi stream (also called the Guikou Stream) in Yongan City, Fujian Province (Figure 1a). The stream is 47 km long and the catchment area is 505 km$^2$. The Baxi stream flows into the Shaxi stream after joining the Jiulong and Ho streams (Figure 1b). The study area focuses on a 6.5 km-long downstream section of the Baxi stream, between the Chen–Nan boulevard and Wu–Yi Road, which passes by the municipal government buildings on the left bank (Figure 1c). Along the streamside of the study area are roads, schools, residences, commercial buildings, green spaces, and landscaped leisure trails. This area is an important hub in Yongan City for human activities, shopping, business, leisure, and sports.

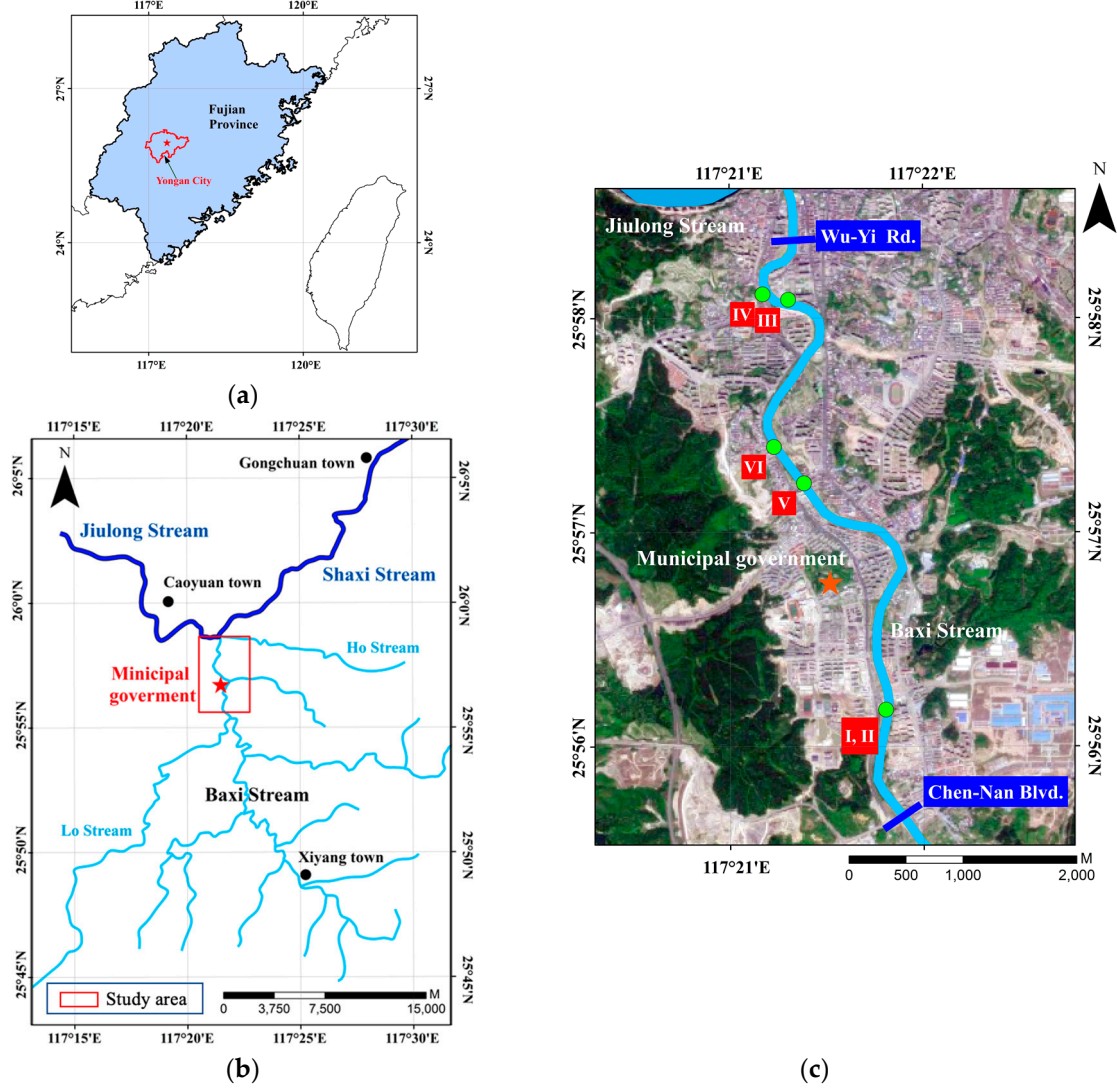

**Figure 1.** The study area of the Baxi stream (also called Guikou Stream) in Yongan City, Fujian Province, China. (**a**) Location of Yongan City in Fujian Province, China. (**b**) The Baxi stream watershed and study area. (**c**) Investigated reach in the 6.5 km-long downstream section of the Baxi, between Chen–Nan Blvd. and Wu–Yi Rd., passing by the municipal government buildings on the left bank. Six images were collected from sites I, II, III, IV, V, and VI (See Figure 2).

The width of the stream in the study area is 40–62 m, and the average gradient of the river is 1/500. According to Strahler's classification of stream order [52], the portion of Baxi stream in the study area is a third-order stream. Hydraulic structures in the studied river reach mainly include revetments (mostly concrete) and submerged dams. Some revetments are paved with pebbles or block stones. However, the slope of the revetments is close to vertical and is not conducive to hosting natural vegetation. Therefore, these revetments cause a visual discontinuity between the riverbank and the riverbed. In addition to the visual impact, the revetments also affect the ecology and water quality, impacting the overall water environment. Four submerged dams have been constructed in the study area to raise the water level, intended to extract water for hydropower generation and to increase the water volume in the riverbed. An increase in the water volume in the riverbed is intended to improve the visual quality of the stream landscape, particularly during the dry season. Among the four submerged dams, two are water-filled rubber dams. Rubber dams are hydraulic structures that can be both inflated and deflated. They have been installed worldwide for various purposes, such as irrigation, water supply, power

generation, tidal barriers, flood control, environmental improvement, and recreation [53]. A rubber dam filled with water can raise the water level in the dry season and release floodwater from the watercourse during the wet season. Consequently, the water level in the studied river reach can be regulated by human intervention, which can consequently change the river landscape. During field surveys, we observed that when the water level rises, the riverbed vegetation disappears (plants are submerged by water); when the water level drops, the riverbed water volume decreases, bare land appears, and vegetation can gradually populate the riverbed. The soil, sand, rock, and vegetation accumulated in the riverbed are often cleared for flood control, which produces a landscape with a flat riverbed and no greenery. Consequently, the landscape elements of water and greenery alternate, which is a common phenomenon in the stream reach of this study.

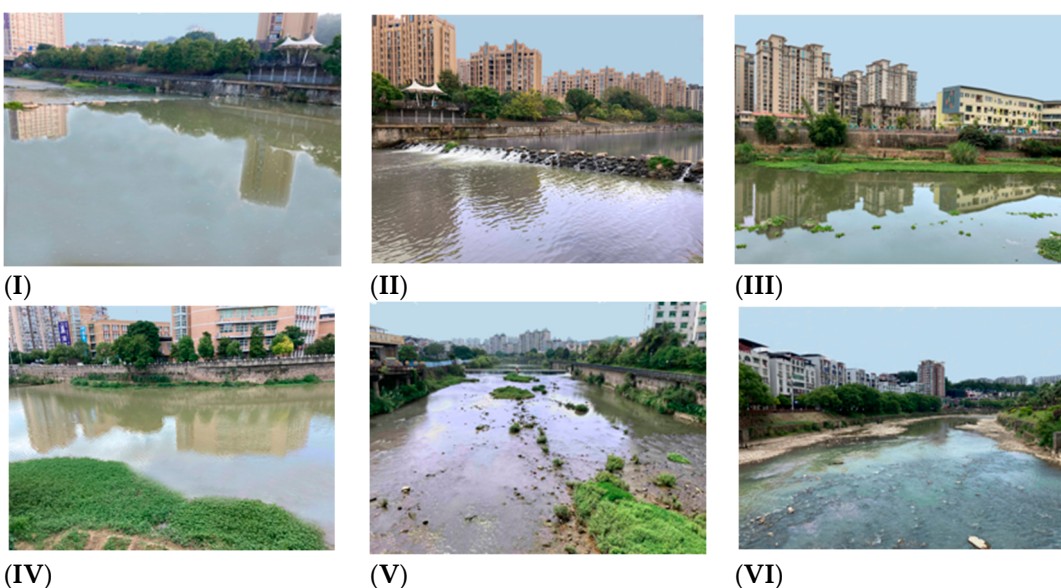

**Figure 2.** Six images from locations (**I**–**VI**), along the Baxi stream. (**I,II**) A cross-river view taken from the rubble dam upstream to downstream and the rubble dam downstream to upstream, respectively. (**III**) A cross-river view taken from upstream of the dam affected by backwater. Most of the floating vegetation is distributed toward the right bank. (**IV**) A cross-river view taken from upstream of the dam affected by backwater, showing that most of the emergent plants are distributed on the left bank. (**V**) A flow-axis view taken from the dam downstream when the rubber dam is raised. (**VI**) A flow-axis view taken from the rubber dam upstream when the dam is lowered.

## 3. Materials and Methods

### 3.1. Collection of Images

We collected images of six locations in the study area (Figure 1c). The chosen locations in the riverbed landscape (Figure 2) are subject to particularly high levels of human intervention, particularly the vertical revetment on the waterfront side that restricts the flow within a fixed range. There are four submerged dams that regulate the water level. Figure 2I,II show the impacts of the rubble dam, which has a fixed height and does not control the water level. Figure 2III,IV show the impacts of the concrete dam, which is used for storing water for power generation. The concrete dam also has gates to regulate the water level. Figure 2V,VI (show the impacts of adjustable dams that can be raised to retain water and lowered to release water. The four submerged dams in the study area can regulate the water level and change the visible volume of water. This study focused on the landscape of the water surface and riverbank and therefore, the area of riverbed and riverbank captured in the images was >50%.

The six images were taken from a naked-eye viewing angle. They also reflected different characteristics of the stream landscape as observed from different perspectives,

capturing different landscape elements in the stream. Images are taken from two key perspectives: (1) the observer stands on the river bank to observe the landscape on the opposite bank, so the view is of a single bank (i.e., left or right bank), called a cross-river view (Figure 2I–IV); and (2) the observer stands on the Xi-Hua bridge to observe the upstream or downstream landscape, and the view covers both banks. This is called a flow-axis view (Figure 2V,VI). According to the landscape elements on the riverbed, we determined three main types of water surface: a single water surface (Figure 2I,VI); a water surface with vegetation (Figure 2III–V); and a water surface with submerged dams (Figure 2II). Using the images shown in Figure 2, we adjusted the proportions of greenery and water on the riverbed by image processing using Adobe Photoshop. Figure A1 in the Appendix A shows the images used for the questionnaire survey. Six groups of photos were presented based on Figure 2I–VI. Each group had four images showing changes in the stream landscape due to variations in water and vegetation.

### 3.2. Questionnaire Survey

We conducted a survey to understand the cognitive levels of respondents in each group of images (Figure A1). To assess the cognitive level, we first chose cognitive indicators and then rated the score of each cognitive indicator. We selected four cognitive indicators: harmony, naturalness, vividness, and preference. These cognitive indicators have been demonstrated to be helpful for evaluating the visual quality of sediment control structures in mountain streams [17,29] and therefore were used in the present study. A five-point Likert scale was used to rate each of these descriptors (harmony, naturalness, vividness, and preference) as "very high", "high", "medium", "low", or "very low". The corresponding scores were 1, 0.8, 0.6, 0.4, and 0.2, respectively, and were assessed in a subsequent quantitative analysis. A hard copy of the survey was sent to each respondent and was collected after completion.

The survey was issued to teachers and students at Fujian College of Water Conservancy and Electric Power in Yongan City, Fujian Province. Most students (>90%) and teachers (>70%) from the college were born in Fujian Province. Even teachers from outside the province have lived in Fujian Province for a long time and are already familiar with the provincial environment. We suggest, therefore, that most of the survey respondents are familiar with the landforms, climate, and food culture of Fujian Province. Furthermore, since the study site is adjacent to the campus, the respondents have experience passing through the study area. People with different socio-cultural backgrounds may have significant differences in how they use and perceive natural landscapes [54], and there are many factors affecting socio-cultural backgrounds, such as length of residence, socio-demographic characteristics, knowledge, beliefs and values, and place attachment [21]. Therefore, the respondents were divided into two categories (experts and general public) according to differences in their background knowledge. The expert group comprised people with academic backgrounds (graduate degrees) or occupations related to river planning and design, including hydraulic engineering, civil engineering, landscape engineering, landscape architecture, or environmental design. The general public group comprised professionals other than those in the expert group. The general public group was mainly freshmen and sophomores, all of whom had a high school education or above.

### 3.3. Quantitative Analyses

We quantified two key components in this study. The first quantifiable component was the survey results, compiled by assigning scores from 0.2–1.0 for each Likert grade. We then statistically analyzed the survey scores, determined the average values, and analyzed the correlation between indicators.

The second quantifiable component was the percentage of the main landscape elements in each image. The landscape elements in the images included water, plants, bare land, buildings, and infrastructures, such as submerged dams, revetments, and other public facilities. A high proportion of each image in this study comprised water, greenery, build-

ings, and infrastructure. To assess the visible landscape, we determined ratios of visible water ($W_R$), visible greenery ($G_R$), visible buildings ($B_R$), and visible infrastructure ($I_R$), which were calculated as follows:

$$W_R = W/T \tag{1}$$

$$G_R = G/T \tag{2}$$

$$B_R = B/T \tag{3}$$

$$I_R = I/T \tag{4}$$

where $T$ is the total number of pixels in an image, and $W$, $G$, $B$, and $I$ are the number of image pixels of water, greenery, buildings, and infrastructure, respectively. The number of pixels was determined using Adobe Photoshop.

## 4. Results and Discussion

Responses were received from 56 expert surveys and 261 general public surveys. The effective questionnaire of the expert group was 87.5%, and that of the general public group was 85.8%. The age range of respondents in the expert group was 20–60 years old, of which 87.8% of respondents were aged 20–40 years old. The age range of respondents in the general public group was 18–40 years old and 58.9% of respondents were aged 18–20 years old (Table 1). We conducted a reliability test to check the internal consistency of the survey data using the Cronbach coefficient $\alpha$ [55]. The $\alpha$ coefficient in this study ranged from 0.764 to 0.989, confirming that the data collected shared relatively high interclass reliability. The validity of four cognitive indicators ($N$, $V$, $H$ and $P$) was confirmed using a construct validity test via Pearson's correlation coefficient $r$. The coefficient $r$ for all indicators was between 0.550 and 0.593, all with a significance of $p < 0.001$, indicating that the validity coefficient in this research questionnaire is higher than the criterion of $r = 0.45$ [56], and the validity is very beneficial ($r > 0.35$) [57,58].

**Table 1.** Background data of survey respondents.

| Types of Respondents | Received Samples, $n$ | Effective Samples, $n_e$ | $n/n_e$ (%) | Gender Percentage in $n_e$ (%) | | Age Percentage in $n_e$ (%) | | | |
|---|---|---|---|---|---|---|---|---|---|
| | | | | Male | Female | 18–20 | 20–40 | 40–60 | >60 |
| General public | 261 | 224 | 85.8 | 73.2 | 26.8 | 58.9 | 41.1 | 0 | 0 |
| Expert | 56 | 49 | 87.5 | 38.8 | 61.2 | 0 | 87.8 | 12.2 | 0 |

### 4.1. Preference Associated with Cognitive Indicators

To consider the degree of preference, harmony, vividness, and naturalness, we determined the mean values of $P$, $H$, $V$, and $N$. Figure 3 shows preference associated with cognitive indicators for the expert and general public groups for the different image sets (I, II, III, IV, V, and VI). We found no substantial difference between the image sets and the different questionnaire groups (Figure 3). $P$ has a positive correlation with indicators $H$, $V$, and $N$, with correlation coefficients of $r = 0.966$, $0.895$, and $0.908$, respectively, with significance values of $p < 0.001$. We also found that $V$ and $N$ are highly correlated with each other, with $r = 0.972$ at $p < 0.001$.

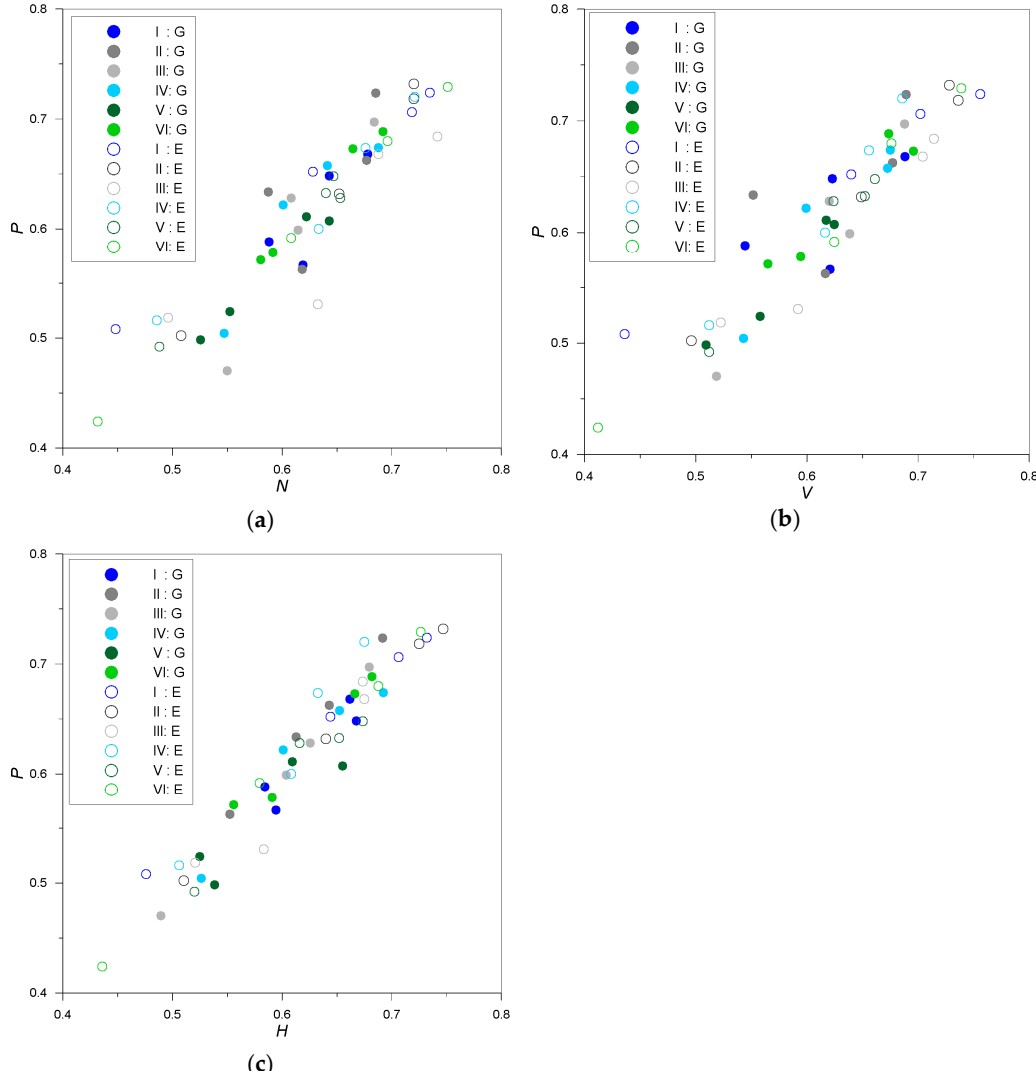

**Figure 3.** Relationship between preference *P* and cognitive indicators (naturalness *N*, vividness *V*, and harmony *H*) for the expert (E) and general public (G) groups for different image sets (I, II, III, IV, V, and VI). (**a**) Preference *P* and naturalness *N* ($r$ = 0.908 at $p$ < 0.001 when two groups, G and E, were analyzed together). (**b**) Preference *P* and vividness *V* ($r$ = 0.895 at $p$ < 0.001 when two groups, G and E, were analyzed together). (**c**) Preference *P* and harmony *H* ($r$ = 0.966 at $p$ < 0.001 when two groups, G and E, were analyzed together).

We found that people prefer stream landscapes that are visually harmonious, vivid, and natural (Figure 3). In particular, the correlation between harmony *H* and preference *P* is very high (Figure 4). Notably, we found a similar response for *P* and *H* between different questionnaire groups (i.e., experts and the general public), so we did not distinguish between the two groups for statistical analysis. *P* is almost consistent with *H*, which can be expressed as *P* = *H* with $r$ = 0.966. Therefore, we found that preference is equivalent to harmony, and *P* and *H* are often used interchangeably [29,40]. Harmony was used to replace preference in a study of the visual quality of sediment control structures in mountain streams [29], and the results were comparable to those found here (Figure 4). Our study does not distinguish between different sample groups (experts and the general public) in the following analysis but averages of all samples for each image.

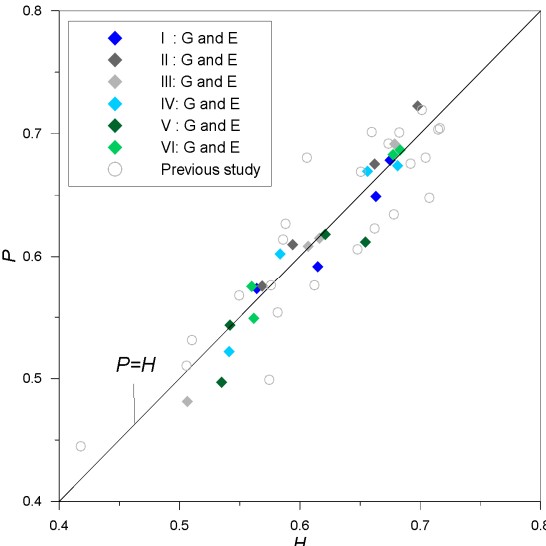

**Figure 4.** Fitted line between preference *P* and harmony *H* for all images in this study and compared with previous study [29] Note that we do not distinguish *P* and *H* values between different survey respondent groups (expert, E, and general public, G) but we instead average all survey data for each image.

### 4.2. Harmony Associated with Landscape Elements

Landscape elements include water bodies, plants, vegetation, buildings, and hydraulic facilities. Harmony or coherence has been associated with water features [38,59,60], proportion of vegetation [29], properties of engineering structures (i.e., scale and texture) [29], and land use suitability [38,61,62]. In this study, we evaluated the visual area of all elements in the image and determined the $G_R$, $W_R$, $B_R$, and $I_R$ indicators. We also analyzed the correlation of these indicators with harmony.

#### 4.2.1. Relationship between Harmony and Visible Water

Figure 5a,b illustrate the relationship between *H* and $W_R$ for the six locations (I–VI) and two perspectives (cross-river view and flow-axis view), respectively. The relationship between *H* and $W_R$ is independent of location and perspective. All results show that harmony *H* gradually increases and then gradually decreases with increasing amounts of visible water $W_R$. In addition, artificial elements can adversely affect the visual satisfaction of an observer [63]. Here, we discuss the impact of the proportion of buildings and facilities on harmony. We found that the influence of the proportion of buildings and facilities on harmony is not significant (Figure 5c,d); this may be because of a low proportion of buildings and facilities ($B_R + I_R < 25\%$). If the visual proportion of mountain-stream engineering structures in an image is high (>30%), visual harmony can be affected [29]. Harmony peaks at a 32% of visible water $W_R$, and too much or too little $W_R$ reduces the visual harmony of the river landscape (Figure 6). The empirical relationship between harmony (*H*) and the ratio of visible water ($W_R$) can be expressed as

$$H = -1.69(W_R - 0.32)^2 + 0.66, \tag{5}$$

which is obtained by fitting the data, where the coefficient of correlation is *r* = 0.71.

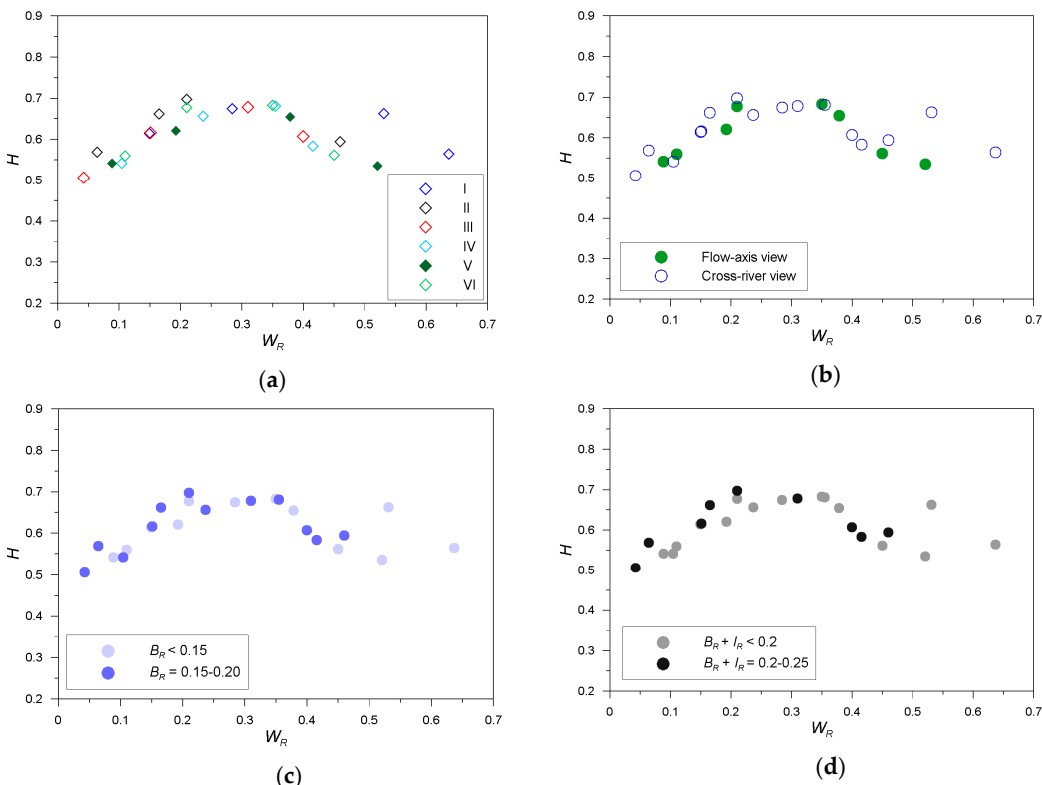

**Figure 5.** Relationship between harmony (*H*) and amount of visible water ($W_R$) for different sites and views, for various values of the ratio of visible buildings $B_R$, and the proportion of visible buildings and infrastructure ($B_R + I_R$). (**a**) Different sites. (**b**) Different views. (**c**) Various values of proportion of visible buildings $B_R$. (**d**) Various values of proportion of visible buildings and infrastructure ($B_R$ and $I_R$).

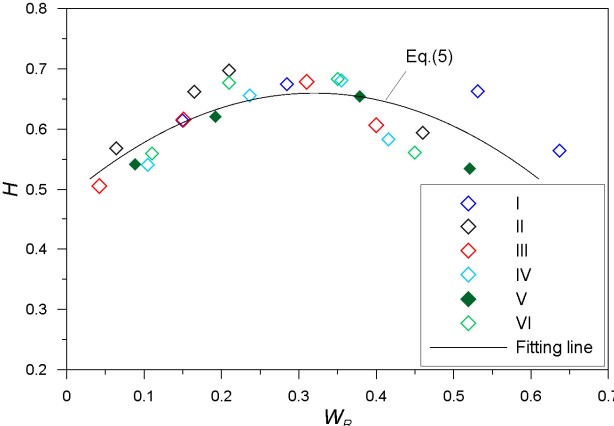

**Figure 6.** Empirical relationship between harmony (*H*) and amount of visible water ($W_R$).

### 4.2.2. Relationship between Harmony and Visible Greenery

Following the same procedures as in the previous section, we analyzed the relationship between harmony (*H*) and the ratio of visible greenery ($G_R$) at different sites (Figure 7a) and views (Figure 7b) for various values of $B_R$ and $B_R + I_R$ (Figure 7c,d). Similar to the $H–W_R$ relationship, the $H–G_R$ relationship was not significantly affected by site or perspective. *H* increased and then decreased with increasing $G_R$. *H* peaked when $G_R$ was in the range of

30–50%. Based on the $H$–$G_R$ relationship identified here, the best-fit line with a correlation coefficient of $r = 0.65$ (Figure 8 and represented by the blue solid line) can be expressed as:

$$H = -1.36(G_R - 0.37)^2 + 0.65. \tag{6}$$

Similar data and equations have been proposed by previous studies on the visual harmony of hydraulic structures in mountain streams [29]. The previous empirical relationship was expressed as:

$$H = -1.45(G_R - 0.5)^2 + 0.7. \tag{7}$$

Equation (7) is indicated in Figure 8 and represented by the black solid line. The two bounding curves in Figure 8 are obtained from the data for hydraulic structures in mountain streams [29] and are presented for comparison with the present study. The data in the present study fall between these two bounding curves. We found the ratio of visible greenery at peak harmony for an urban stream ($G_R = 0.37$, $H = 0.65$) to be smaller than that in the previous study on mountain streams ($G_R = 0.5$, $H = 0.7$). This inconsistency may be due to differences in the topographic features and compositions of landscape elements between urban and mountain streams (Table 2). Most mountain streams investigated by Chen et al. [29] could be classified as first- to second-order streams, while the urban stream in the study area is a third-order stream. Generally, the visible greenery coverage is lower, and the amount of visible water is higher in the urban stream than in the mountain stream, and there are different types of hydraulic structures.

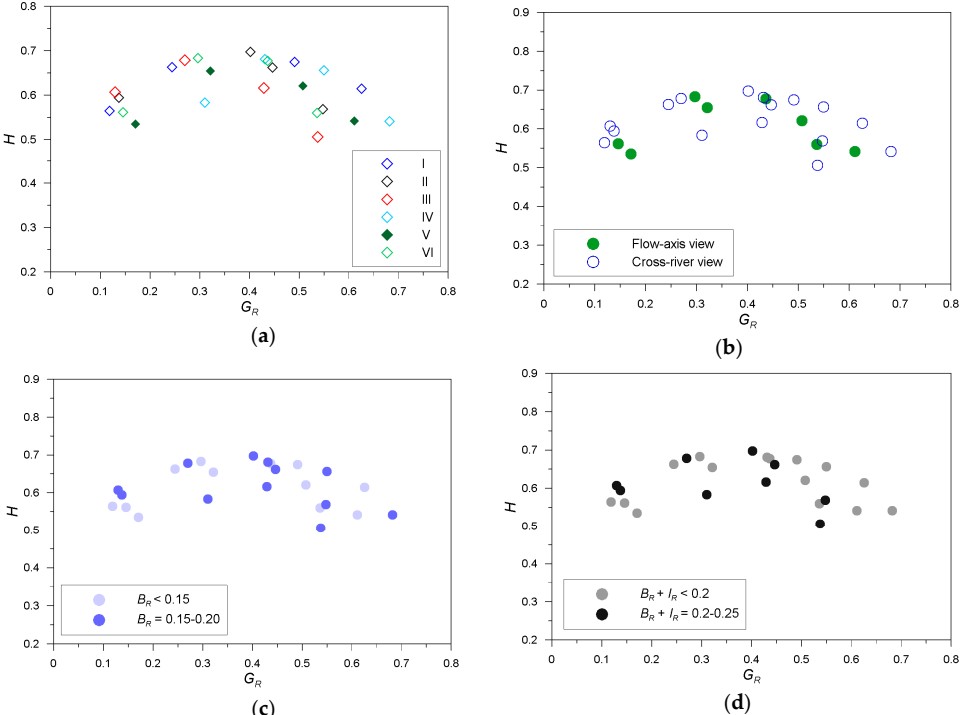

**Figure 7.** Relationship between harmony (*H*) and the ratio of visible greenery (*G_R*) for different sites and views, at various values of the ratio of visible buildings $B_R$, and the proportion of visible buildings and infrastructure ($B_R + I_R$). (**a**) Different sites. (**b**) Different views. (**c**) Various values of proportion of visible buildings $B_R$. (**d**) Various values of proportion of visible buildings and infrastructure ($B_R + I_R$).

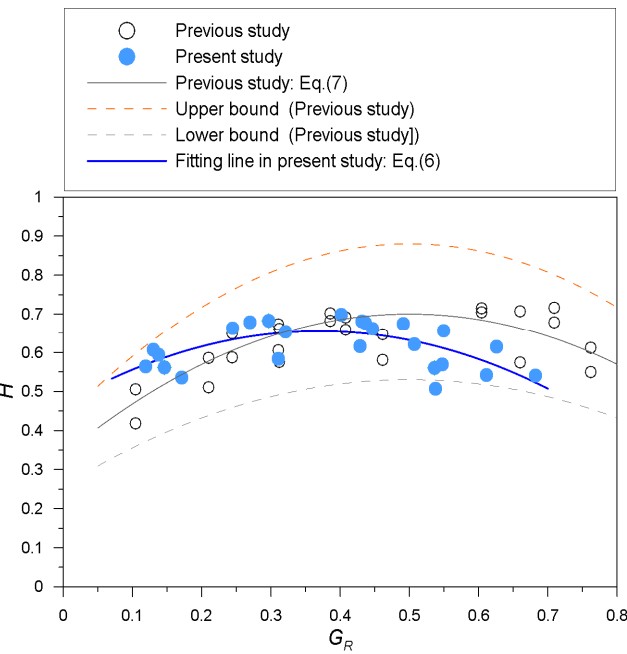

**Figure 8.** Empirical relationship between harmony (*H*) and the ratio of visible greenery ($G_R$) and comparison with previous study [29].

**Table 2.** Comparison of mountain and urban streams.

| Study Site | Stream Order * | Topographical Features | Landscape Elements | Common Types of Hydraulic Structures | Parameters of Visible Landscape Element (%) | Sources |
|---|---|---|---|---|---|---|
| Mountain streams, Taiwan | 1, 2 | Stream with steep slope (5–40%) and low width (5–20 m) | Water, plants and hydraulic structures | Groundsill, submerged dam, check dam, and revetment | $G_R$ = 5–85; $W_R$ = 0–42; $B_R$ = 0; $I_R < 80$ | Chen et al. [29] |
| Urban section of Baxi stream, Yongan, Fujian Province | 3 | Stream with flat slope (0.2%) and high width (40–62 m) and passing through a city | Water, plants, hydraulic structures, and buildings | Groundsill, submerged dam, weir, and revetment | $G_R$ = 13–68; $W_R$ = 10–63; $B_R$ = 5–18; $I_R$ = 1.8–9.5 | This study |

Note(s): * Based on Strahler's classification [52].

### 4.2.3. Relationship between Harmony and Ratio of Water to Greenery

Urban riversides have spaces for human living and activity, such as houses, footpaths, roadways, and parks or green spaces. To protect these spaces, urban rivers often include revetments or embankments that confine the water flow to protect human life and property. Consequently, within a fixed range, the amount of visible water and greenery (particularly natural vegetation in the riverbed) show a mutual growth and decline trend. An increase in the amount of visible water relates to a decrease in natural vegetation on the riverbed. Therefore, we further explored the proportion of visible water and greenery associated with harmony *H* or preference *P* (Figures 9 and 10). The empirical relationship between *H* and the ratio of water to greenery ($W_R/G_R$) can be expressed as:

$$H = 0.69 \exp\left[-0.51(\log(W_R/G_R) - \log(0.78))^2\right]. \tag{8}$$

Based on the high correlation and consistency between $H$ and $P$ (Figure 4), that is, $P = H$, we replace $H$ with $P$ in Equation (8) to obtain

$$P = 0.69 \exp\left[-0.51(\log(W_R/G_R) - \log(0.78))^2\right].\qquad(9)$$

Equations (8) and (9) have correlation coefficients of $r = 0.837$ and $r = 0.796$, respectively. $W_R/G_R$ against $H$ has a higher $r$ than $W_R/G_R$ against $P$. Equation (8) shows the highest $H$ at $W_R/G_R = 0.78$. $H$ increases when $W_R/G_R < 0.78$, and $H$ decreases when $W_R/G_R > 0.78$. Therefore, the optimal ratio of visible water to visible greenery ($W_R/G_R$) is approximately 0.8. Deviation from this value reduces visual harmony and preference. When the value of $W_R/G_R$ is too low, a moderate increase in the amount of visible water $W_R$ will improve visual harmony and preference. However, only increasing the visible water amount without regulating visible greenery reduces the harmony and preference. For many submerged dams built in the study area (Figure 2I,II), the purpose is to increase the amount of visible water, thus improving the visual quality and increasing public perception. However, because of the lack of regulation of visible greenery, the original purpose is not always achieved. If the green strip at the riverside can be appropriately increased or if the value of $W_R/G_R$ is adjusted to approximately 0.8 under flood control safety, then the harmony of the riverside landscape or the visual preference will improve.

Increasing riparian vegetation not only protects the riverbank from the direct impact of a flood but also maintains biodiversity in the river area [5]. There are several methods to adjust the water-to-greenery ratio in terms of engineering technology. These are: (1) add soil or sediment to the waterfront side of the revetment foundation to protect the sloped surface with block stones so that plants can naturally inhabit the fill area under stable conditions, which can increase the visible greenery; (2) in still water, such as upstream of submerged dams or in areas with slow flow velocity, ecological floating bed technology [64] can be used to increase the visible greenery at the water surface; and (3) the water retention height of the submerged dam can be adjusted according to the terrain or riverbed preparation and controlled at a more optimum value of $W_R/G_R$ (approximately 0.8). These engineering technology methods could be used to improve the visual harmony and river landscape beauty in the study area. However, these methods would require an accurate hydraulic analysis to confirm flood control safety.

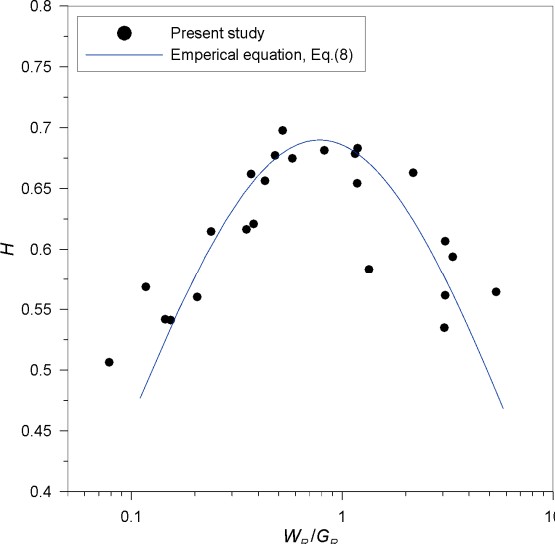

**Figure 9.** Relationship between harmony ($H$) and ratio of water to greenery ($W_R/G_R$).

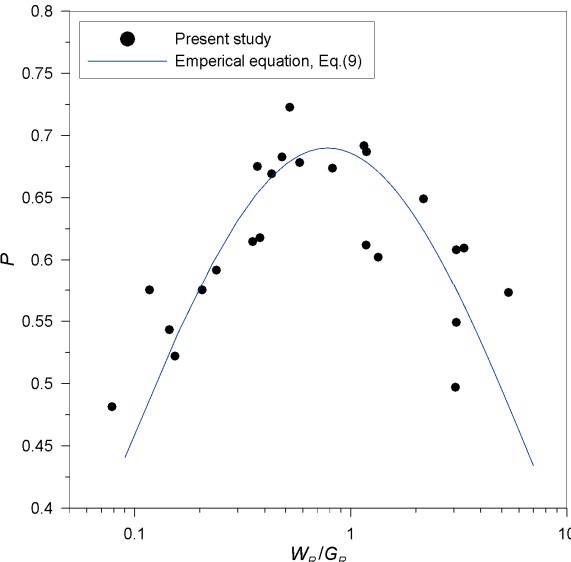

**Figure 10.** Relationship between visual preference ($P$) and ratio of water to greenery ($W_R/G_R$).

### 4.3. Suitability of H–$W_R/G_R$ and P–$W_R/G_R$ Relationships

The empirical Equations (8) and (9) for harmony and preference against the ratio of water to greenery were obtained by changing the amounts of greenery and water at the waterfront. These equations could be more suitable for assessing the harmony or public favor of a river landscape because of the changes in greenery and water amounts in the study area. Therefore, we collected images from the stream reach of the study area (Figure A2) and distributed a second survey to check the suitability of the proposed equations. A total of 185 surveys were sent and 157 valid survey responses were received. We conducted a statistical analysis of the relevant parameters following the methods described in Section 3. The relationship between expected harmony (calculated using the empirical equation) and harmony obtained from the survey results is shown in Figure 11. Following the same process shown in Figure 11, $H$ was replaced with $P$ (Figure 12). The results are similar (Figures 11 and 12). They indicate that the calculated harmony or preference agrees with the survey harmony or preference from the data of Figure A1; that is, most data follow the solid line in Figure 11, while some data from Figure A2 diverge away from the line, such as images 1, 5, and 6. These data points (1, 5, and 6) show an underestimation of calculated harmony or preference.

Among the 11 images in Figure A2, images 5 and 6 have a high degree of artificial facilities ($I_R + B_R > 20\%$), while image 1 has less human intervention and is closer to its natural state. Therefore, images with more artificial facilities or closer to their natural state may not be relevant for use with Equations (8) and (9). The results of the empirical equation obtained from the images in Figure A1 all have the same background color (a single sky color) and show obvious human interference. In particular, stream flow was confined by a vertical revetment and influenced by a submerged dam. Therefore, Equations (8) and (9) may not be applicable to images of a stream landscape with changes to the background sky color. For example, many clouds are present in the sky of image 5. In addition, a high proportion and variety of artificial facilities are visible, such as white sidewalk guardrails in image 5 and brown wooden handrails in image 6. Images 5 and 6 imply that interventions of appropriate colors or materials may help to improve the vividness or harmony of the landscape and may also enhance visual preference. Furthermore, the color (e.g., water color), size, arrangement, and distribution of landscape elements may affect the reasonability of Equations (8) and (9).

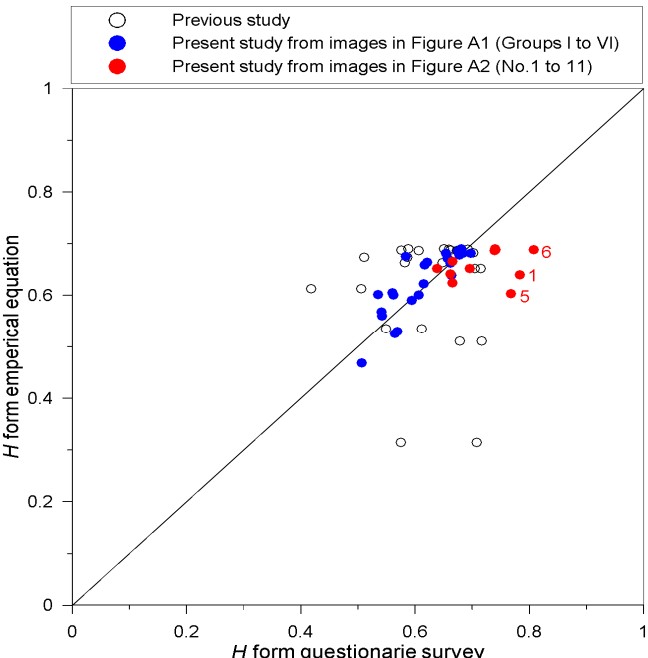

**Figure 11.** Relationship between the visual preference *H* from the empirical Equation (8) and *H* from the questionnaire survey. Data from the present study of a third-order urban stream and published results [29] for first- to second-order mountain streams. Images 5 and 6 have a high degree of artificial facilities ($I_R + B_R > 20\%$), while image 1 has less human intervention and is closer to its natural state.

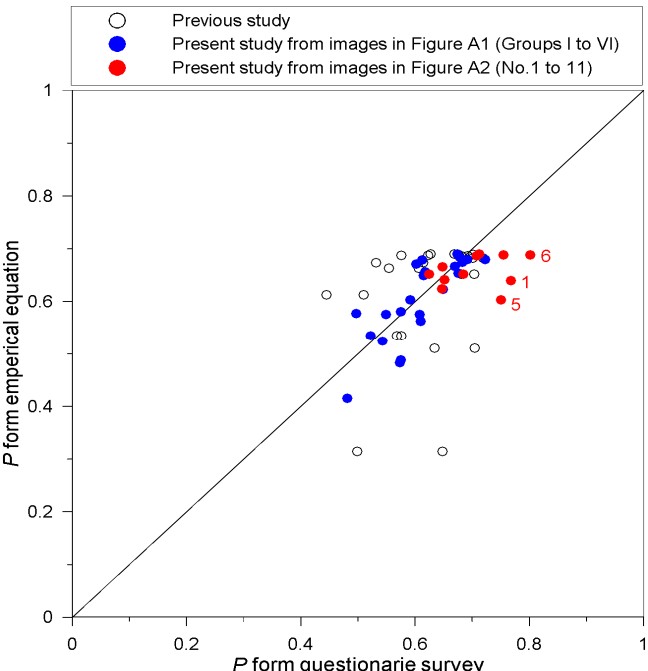

**Figure 12.** Relationship between the visual preference *P* from the empirical Equation (9) and *P* from the questionnaire survey. Data from the present study of a third-order urban stream and published results [29] for first- to second-order mountain streams. Images 5 and 6 have a high degree of artificial facilities ($I_R + B_R > 20\%$), while image 1 has less human intervention and is closer to its natural state.

Figures 11 and 12 also compare our results with those from a previous study [29], to demonstrate that the proposed equations in this study (Equations (8) and (9)) are not applicable to mountain streams with hydraulic structures. This is because of the different topographical characteristics and features of landscape elements between mountain and

urban streams. Chen et al. [29] focused on the landscape of mountain streams with sediment control structures. These streams were first- to second-order streams and were located in the headwater or upstream. However, the characteristics or elements of stream landscapes in steep mountainous areas differ from those in gently sloping urban areas. For example, from a visual perspective, there is usually less visible water and more greenery in mountain streams than in urban streams and there are fewer or no buildings in mountain streams, whereas there are many buildings visible from urban streams. Hydraulic structures also differ, where tall check dams are commonly used to control sediment in mountain streams, but rarely in urban streams.

Our study assesses the visual harmony of riparian landscapes due to changes in visible water and greenery, considering urban streams that are affected by hydraulic structures. Our results could provide a reference for improving the visual beauty or harmony in the planning and design of the Baxi stream. For example, in the case of Figure 2I (or Figure A1(I1)) with $H = 0.56$, $G_R = 0.12$, $W_R = 0.64$, and $W_R/G_R = 5.33$, the visible greenery, visible water, and the ratio of water to greenery is inappropriate. Improved values would be $G_R = 0.37$, $W_R = 0.32$, and $W_R/G_R = 0.78$. The river landscape can be improved by adjusting the proportion of water and greenery using engineering techniques, such as the construction of a green strip along the river, introducing an ecological floating bed on the water surface, or adjusting the dam height.

## 5. Conclusions

The Baxi stream in China was selected as an example site to study the visual harmony of the relative proportions of water and greenery in urban streams. The stream landscape was affected by a vertical revetment and submerged dams, and the visible amount of water and vegetation varied. The conclusions of our study are summarized as follows:

(1) Visual preference $P$ is positively correlated with harmony $H$, vividness $V$, and naturalness $N$. In particular, $H$ is almost equal to $P$. Visual harmony or visual preference was affected by the ratio of visible water amount $W_R$, the ratio of visible greenery $G_R$, and the ratio of $W_R$ to $G_R$. We presented empirical equations for $H$ or $P$ with $W_R$ and $G_R$. Visual harmony or preference is optimal at $W_R$ values of 0.2–0.4 (peak harmony at $W_R = 0.32$) and $G_R$ values of 0.3–0.5 (peak harmony at $W_R = 0.37$). The optimal value of $G_R$ in this study of urban streams is lower than that determined for mountain streams.

(2) We proposed an empirical relationship between $H$ or $P$ and the water-to-greenery ratio ($W_R/G_R$). $H$ increases when $W_R/G_R < 0.78$ and $H$ decreases when $W_R/G_R > 0.78$. $H$ or $P$ was optimal at $W_R/G_R = 0.78$. This value is important for improving the visual quality of stream landscapes in the study area. Many submerged dams have been constructed in the study area to increase the amount of visible water. However, because of the lack of regulation of visible greenery, the original purpose was not achieved. Under flood control conditions, when the green zone of the riverbed is appropriately increased, a $W_R/G_R$ value of approximately 0.8 will improve the harmony of the riverside landscape or improve visual preference.

(3) This study used image processing to simulate images of the river landscape with changes in the water-to-greenery ratio ($W_R/G_R$). We then analyzed these images through surveys to study the visual harmony of the urban stream. We selected cognitive indicators ($P$, $H$, $N$, and $V$) and eco-physical factors (mainly $W_R$ and $G_R$), and proposed a logarithmic normal function of $H$ or $P$ and $W_R/G_R$. This methodology could provide other study areas with a reference for how to obtain the best visual harmony or achieve public acceptance by changing the amount of visible water and/or greenery. This can be achieved via dam construction or changes in hydrological characteristics in urban rivers.

(4) The optimal $W_R/G_R$ in the study area was approximately 0.8. However, the optimal $W_R/G_R$ may be different in different regions because of different landforms, infrastructure, and the socio-cultural background of respondents. The survey respondents in this study were mainly college members who were familiar with the local envi-

ronment in Fujian Province. The socio-cultural differences between the respondents were not significant. Different sociocultural factors, such as length of residence, socio-demographic profile, knowledge, beliefs and values, and place attachment [21] may have an impact on the visual harmony and preference of river landscapes, which still needs further research.

(5) The empirical equations proposed in this study were obtained from images with river waterfront coverage exceeding 50%, where the waterfront appearance was disturbed by vertical revetments and submerged dams. The images included vegetation, water, and sky with no substantial color changes. An image with most buildings and infrastructure may not be a valid input for the equation presented here. Subsequent research should collect images of similar rivers in other cities to verify the rationality of the empirical equations. This study discusses third-order urban streams. The proportion of landscape elements for different river orders could differ, and the relationship between the harmony of the river landscape and landscape elements, therefore, requires further investigation.

**Author Contributions:** Conceptualization, J.-C.C.; methodology, J.-C.C.; Field investigation and questionnaire survey, X.-R.F., X.-Z.L., G.-L.L., J.-Q.F., F.-B.L. and J.-C.C.; Image processing and quantitative analyses, X.-R.F. and J.-C.C.; Statistical analysis, J.-C.C., J.-Q.F., X.-R.F., F.-B.L., X.-Z.L. and G.-L.L.; writing and editing, J.-C.C.; funding acquisition, J.-C.C. All authors have read and agreed to the published version of the manuscript.

**Funding:** This research was funded by the Scientific Research Fund of Fujian College of Water Conservancy and Electric Power, Grant Number YJRCKYQD2101, and Sanming City Science and Technology Bureau, China, Grant Number 2020-S-81.

**Data Availability Statement:** The datasets generated and/or analyzed during the current study are available from the corresponding author on reasonable request.

**Conflicts of Interest:** The authors declare no conflict of interest. The funders had no role in the design of the study; in the collection, analyses, or interpretation of data; in the writing of the manuscript, or in the decision to publish the results.

## Appendix A

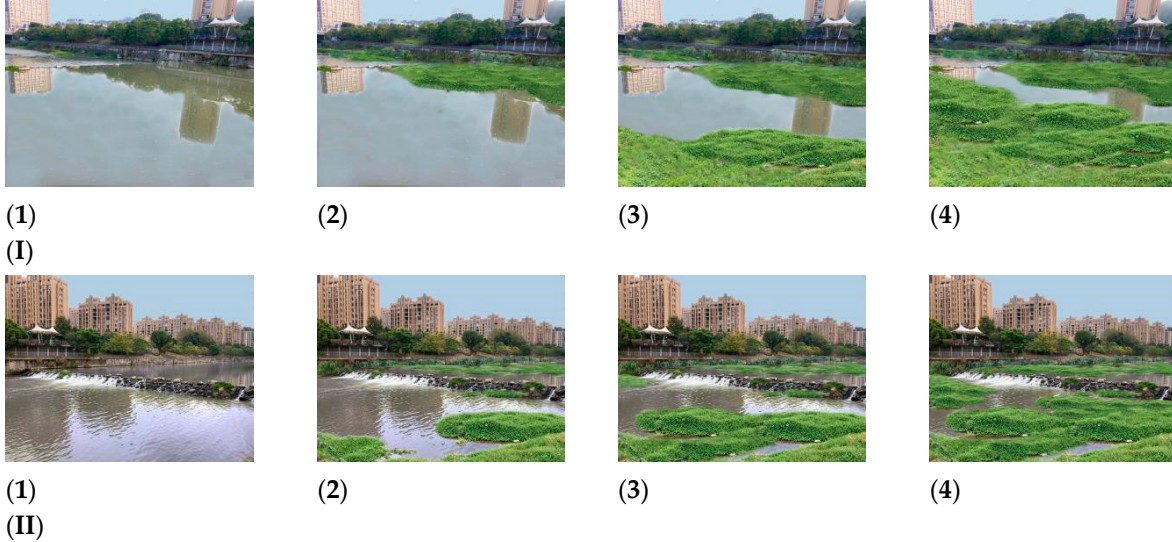

**(1)** **(2)** **(3)** **(4)**
**(I)**

**(1)** **(2)** **(3)** **(4)**
**(II)**

**Figure A1.** *Cont.*

**(1)** **(2)** **(3)** **(4)**

**(III)**

**(1)** **(2)** **(3)** **(4)**

**(IV)**

**(1)** **(2)** **(3)** **(4)**

**(V)**

**(1)** **(2)** **(3)** **(4)**

**(VI)**

**Figure A1.** Six groups of images (**I–VI**) collected in this study, showing stream landscape variations due to changes in vegetation and water. (**I,II**) A cross-river view taken from the rubble dam upstream to downstream and the rubble dam downstream to upstream, respectively. (**III**) A cross-river view taken from upstream of the dam affected by backwater. (**IV**) A cross-river view taken from upstream of the dam affected by backwater. (**V**) A flow-axis view taken from the dam downstream when the rubber dam is raised. (**VI**) A flow-axis view taken from the rubber dam up-stream when the dam is lowered. Each (**1–4**) in the six groups showing the increase of visual greenery or decrease of visual water.

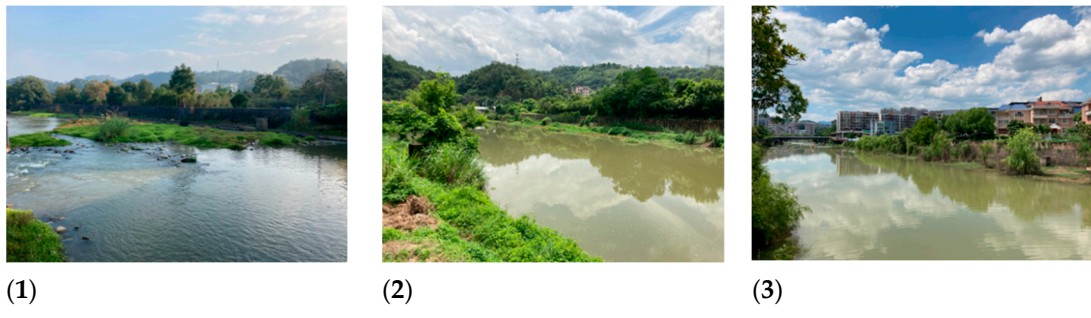

**(1)** **(2)** **(3)**

**Figure A2.** *Cont*.

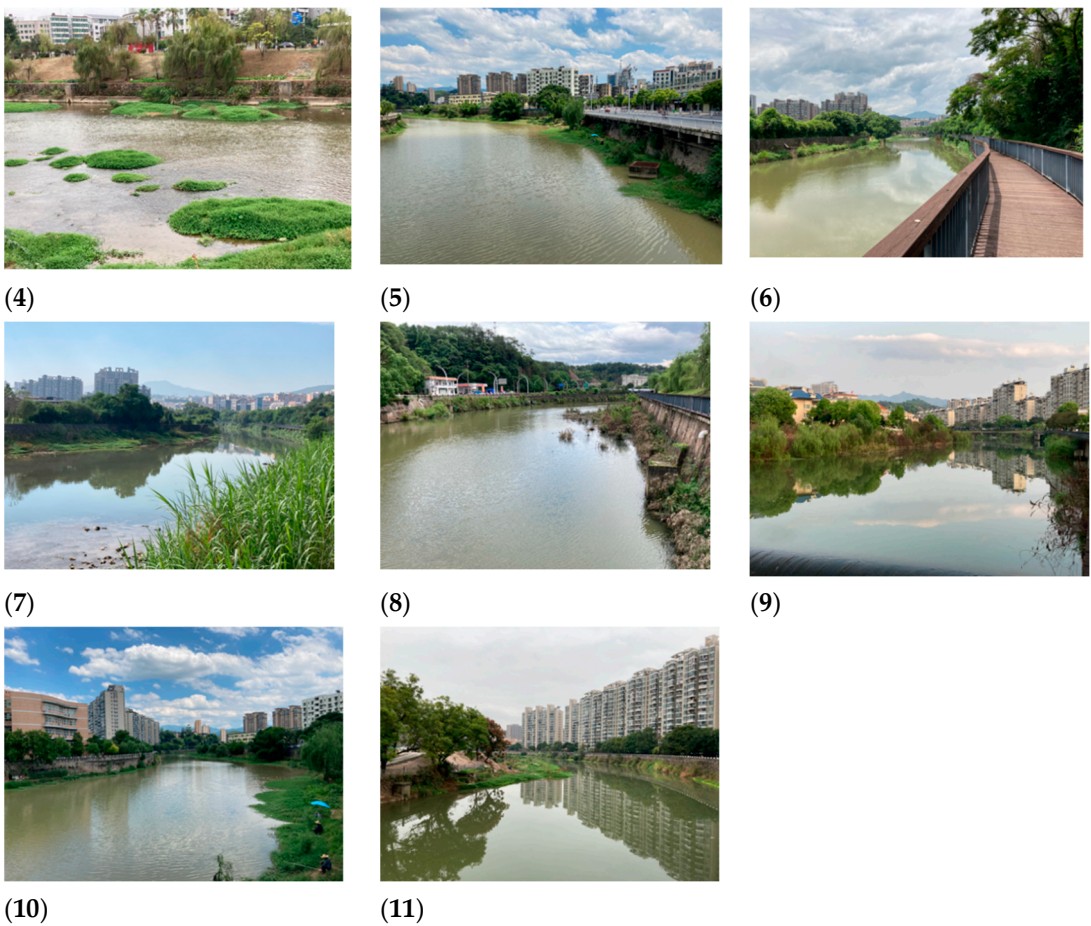

**Figure A2.** Images collected for the second survey on the Baxi stream. (**1,4**) A cross-river view taken from the reach unaffected by submerged dam. (**2,6,7,8,11**) A flow-axis view taken from the reach unaffected by submerged dam. (**3,5,9,10**) A flow-axis view taken from the reach affected by submerged dam. Images (**5,6**) have a high degree of artificial facilities ($I_R + B_R > 20\%$), while image (**1**) has less human intervention and is closer to its natural state.

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
