# Peer review of "Visual Harmony of the Proportion of Water and Greenery in Urban Streams: Baxi Stream, Yongan City, China"

_water, doi:10.3390/w15020341_

Round 1
Reviewer 1 Report
The topic of research is very actual for urban sustainable development. The article is logical and structured good. The methodology of research is explained enough. I would like to suggest authors to enrich literacy and add 5-7 more actual publications in the topic like Chen, J. C., Huang, C. L., Chen, S. C., & Tfwala, S. S. (2021). Visual Harmony of Engineering Structures in a Mountain Stream. Water, 13(23), 3324.
Reviewer 2 Report
This paper mainly studies the harmony degree of Urban River water-green proportion. On the one hand, the paper selected 6 photos, changed the proportion of water and greening, and then through the survey to understand the public preferences. On the other hand, 11 photos of landscape elements including water, plants, buildings and facilities were selected to understand the public's preferences. However, I have more doubts about the rationality of the method, and think that the results are not reliable, the study on the planning and design of rivers does not have practical significance.
1. There has been a great deal of research on landscape preference, but the author did not make a systematic summary.
2. The river is 6.5 km long, but the author has only selected 10 or so photographs. Is it representative enough?
3. The questionnaire needs to pass the reliability test and the validity test to be able to do the further analysis, obviously, the author has not done the related work.
4. In the analysis, only the proportion of landscape elements is calculated, and these indicators are used in the analysis of public preference, which has a big loophole. For example, the appearance, height, and color of a building will affect public preferences; the color and configuration of plants will affect public preferences; and the quality, quantity, and form of water will also affect public preferences. It is clear that the choice of indicators is too simple and the results unreliable.
5. When the water-green ratio is about 0.8, the visual harmony degree is the highest, and the public preference is also higher. Different River landscapes have different characteristics, and this conclusion can not be generalized.
6. Should the correlation between landscape indicators and preferences be analyzed and tested for significance?
7. There are other problems:
1) Figure 1 should have latitude and longitude;
2) Figure 3(c) Preference P and naturalness (harmony?) H;
Reviewer 3 Report
The processing of the topic is based on a complex methodology. The analysis of the topic is developed and implemented in detail. The analysis of the topic is developed and implemented in detail. The subject of the investigation and the case study are properly defined, and this is also clearly indicated in the title.
The study focuses on one location and presents its methodology through a focused case study. The methodology undoubtedly enables universal analysis possibilities, but at the same time, the statements made in the conclusion of the paper are based only on the single on-site case study. The question is, to what extent can these results be generalized? The conclusions are already formulated at an overly general level. It would be important to define the study's limitations and limits of validity. The result of the visual perception and evaluation based on the questionnaire is based on the local cultural arrangement of the local society. That is why the result can be primarily local, and this should definitely be interpreted in the study. The cultural background of aesthetic perception and the limitations of generalizing the results should also be presented.
The graphic quality of the figures should definitely be improved. The content of the map representations is comprehensible, but their appearance is not sophisticated enough. The resolution of the aerial photo is of poor quality.
The paper analyzes the visual quality, but the resolution and size of the images do not allow the reader to make a visual assessment. Better resolution and preferably some larger images would be needed.
Round 2
Reviewer 2 Report
The author made some changes based on the reviewers' comments, but there were minor issues:
1. The results section in the abstract is too small to mention anything but a ratio? so the author need to increase the description of the results;
2. How about the double brackets in the body? (Figure 1(a)) ;
3. In Figure A-1., The V-1 is preceded by what symbol?
